# Development of Fluorescently Labeled SSEA-3, SSEA-4, and Globo-H Glycosphingolipids for Elucidating Molecular Interactions in the Cell Membrane

**DOI:** 10.3390/ijms20246187

**Published:** 2019-12-07

**Authors:** Sachi Asano, Rita Pal, Hide-Nori Tanaka, Akihiro Imamura, Hideharu Ishida, Kenichi G. N. Suzuki, Hiromune Ando

**Affiliations:** 1The United Graduate School of Agricultural Science, Gifu University, 1-1 Yanagido, Gifu 501-1193, Japan; x6103001@edu.gifu-u.ac.jp (S.A.); aimamura@gifu-u.ac.jp (A.I.); ishida@gifu-u.ac.jp (H.I.); 2Department of Applied Bioorganic Chemistry, Gifu University, 1-1 Yanagido, Gifu 501-1193, Japan; rita.ncl@gmail.com; 3Center for Highly Advanced Integration of Nano and Life Sciences (G-CHAIN), Gifu University, 1-1 Yanagido, Gifu 501-1193, Japan

**Keywords:** glycosphingolipid, globo-series, fluorescent analog

## Abstract

Glycosphingolipids (GSLs), such as the globo-series GSLs stage-specific embryonic antigen 3 (SSEA-3), SSEA-4, and Globo-H, are specifically expressed on pluripotent stem cells and cancer cells, and are known to be associated with various biological processes such as cell recognition, cell adhesion, and signal transduction. However, the behavior and biological roles of these GSLs are still unclear. In our previous study, we observed the interactions between the lipid raft and GSLs in real-time using single-molecule imaging, where we successfully synthesized various fluorescent analogs of GSLs (e.g., GM1 and GM3). Here, we have developed fluorescent analogs of SSEA-3, SSEA-4, and Globo-H using chemical synthesis. The biophysical properties of these analogs as raft markers were examined by partitioning giant plasma membrane vesicles from RBL-2H3 cells into detergent-resistant membrane fractions and liquid-ordered/liquid-disordered phases. The results indicated that the analogs were equivalent to native-type GSLs. The analogs could be used to observe the behavior of globo-series GSLs for detailing the structure and biological roles of lipid rafts and GSL-enriched nanodomains during cell differentiation and cell malignancy.

## 1. Introduction

Glycosphingolipids (GSLs) are lipid molecules that are present in the plasma membrane (PM), as well as in cytoplasm and Golgi complex, and contain at least one monosaccharide residue with ceramide lipid that consists of a sphingoid base with a fatty acid amide at the C2 amine [1]. GSLs play significant roles in the PM, such as those in receptors for microbial toxins, mediators of cell adhesion, and modulators of signal transduction [2]. These functions are mediated by cellular microdomains in the PM, such as the lipid raft [3,4] and GSL-enriched microdomains [5]. To detail the distributions, dynamics, and interactions of GSLs with other molecules in the PM, we previously developed chemical methods to synthesize various fluorescent analogs of representative GSLs (e.g., GM1, GM2, GM3, GD1b [6], asialo-GM2, and GalNAc-GD1a [7]) that behave like the native GSLs in the PM. Single-molecule tracking using the fluorescent analogs revealed the specific interactions between GSLs and the lipid raft domain in the PM of living cells [8,9,10]. 

Motivated by these results, we have newly developed the fluorescent probes of stage-specific embryonic antigen-3 (SSEA-3), SSEA-4, and Globo-H in this study (Figure 1), commonly having the GalNAcβ(1,3)Galα(1,4)Galβ(1,4)Glcβ(1,1)Cer structure of globo-series GSLs [11,12]. SSEAs are known as surface markers of human embryonic stem cells (hESCs) [13] and human-induced pluripotent stem cells (hiPSCs) [14]. SSEAs are also implicated in the malignancy of cancers, such as the invasion and metastasis of cancer cells [15,16,17,18,19]. Globo-H is a potential tumor-associated antigen on prostate and breast cancer cells [20]. These GSLs are expected to form microdomains [16,21,22], and are considered to be involved in cell-signaling events during embryogenesis or the malignant progression of tumor cells. To understand the molecular mechanisms of microdomain formation on hESCs, hiPSCs, or cancer cell membranes, we designed and synthesized fluorescently labeled analogs of SSEA-3, SSEA-4, and Globo-H, which could be used for single fluorescent-molecule tracking, and examined their biophysical properties as lipid raft markers.

## 2. Results and Discussion

### 2.1. Chemical Synthesis of Fluorescent Probes

#### 2.1.1. Molecular Design and Synthesis Plan

According to our previous results [8], a highly hydrophilic fluorescent dye (such as ATTO594) was used to label the outermost position of the glycan moiety to maximally retain the biophysical and biochemical properties of the native GSLs. The dye could be selectively conjugated with a terminal amine to furnish amido-linked fluorescent analogs. For the SSEA-3 and Globo-H probes, we used a flexible and hydrophilic spacer to connect the dye at the terminal Gal C3 position, based on the distance from the terminal Gal C3 to Neu C9 of the SSEA-4 probe. All of the syntheses of GSLs were achieved. SSEA-3 was synthesized by Ogawa [23] and Danishefsky [24], and SSEA-4 ganglioside was synthesized by Hasegawa [25] and Schmidt [26]. The synthesis of Globo-H was achieved by Danishefsky [27]. In this study, we exploited the glucosylceramide (GlcCer) cassette approach [28] to connect the oligosaccharide and lipid moieties with high efficiency. The three analogs were synthesized from a GalNAcβ(1,3)Galα(1,4)Gal trisaccharide common unit, which underwent coupling with the terminal glycan moieties and GlcCer. 

#### 2.1.2. Synthesis of the Trisaccharide Common Acceptor

The trisaccharide acceptor was synthesized as depicted in Scheme 1. The Galα(1,4)Gal disaccharide **6** was obtained by di-*tert*-butylsilylene (DTBS)-directed α-galactosylation [29,30]. In this reaction, the formation of an undesirable orthoester was inevitable due to 2-*O*-Bz of **4**. Although a large amount of trifluoromethanesulfonic acid (TfOH; 0.9 equiv.) could minimize orthoester formation, the orthoester compound was inseparable because it had the same Rf value as the target disaccharide **6**. The mixture was advanced to cleavage of the 2,2,2-trichloroethoxycarbonyl (Troc) group to produce the disaccharide **7** together with **5** as the hydrolyzed product of the orthoester, which was successfully separated by silica gel column chromatography. For β-galactosaminylation, we employed a 4,6-*O*-*p*-*tert*-butylbenzylidene (TBBzld)-protected GalNTroc donor **8** [31], which was highly soluble in organic solvent, compared to the benzylidenated analog [28]. The trisaccharide **9** was generated by glycosylation in the presence of NIS-TfOH in CH_2_Cl_2_ at −40 °C, with an excellent yield (94%). It is of note that the TBBzld-protected donor enabled the large-scale synthesis of the carbohydrate building block, owing to its high solubility. The resulting **9** was converted into the trisaccharide common acceptor **10** for the three target molecules by cleavage of the Troc groups, followed by the selective acetylation of the C2 amino group.

#### 2.1.3. Synthesis of the Fluorescent SSEA-3 Probe

For synthesis of the SSEA-3 analog, trisaccharide acceptor **10** was elongated with glycosylation at the 3-OH with a spacer linked to Gal donor **11** to obtain tetrasaccharide **12**, with a yield of 84%. Then, compound **12** was treated with tributylamine hydrofluoride [32] to remove the DTBS group, followed by acetylation, affording compound **13**. The TBBzld and benzyl groups were replaced with acetyl groups to give the fully-acylated derivative **14**. After cleavage of the MP group, an *N*-phenyltrifluoroacetimidoyl group [33] was introduced at the reducing end, providing tetrasaccharide donor **16**.

To construct the glycolipid framework of SSEA-3, we applied a highly soluble GlcCer acceptor **17** as a coupling partner for the tetrasaccharide donor **16**, which was chemically synthesized by glycosylation of 3-*p*-*tert*-butylbenzoyl (TBBz)-protected Cer acceptor with fully protected Glc donor [31]. In GlcCer **17,** bulky TBBz groups were installed as self-aggregation inhibitors, making **17** highly soluble in organic solvents. This easy-to-handle molecule could overcome the solubility issue of GlcCer, which provided a yield of 65% for SSEA-3 skeleton **18**. After global deprotection of **18**, ATTO594 dye was conjugated with the terminal amine to provide ATTO594 SSEA-3 **1** (Scheme 2).

#### 2.1.4. Synthesis of the Fluorescent SSEA-4 Probe

Recently, we developed a method to construct α-sialoside with complete stereoselectivity [34]. In the present study, we exploited this method to construct the glycan moiety of SSEA-4 by sialylation of the SSEA-3 tetrasaccharide.

First, the common trisaccharide acceptor **10** was coupled with the 3,4-*O*-levulinoyl-protected galactosyl donor **21,** affording the tetrasaccharyl derivative **22** with a yield of 87%. Then, the levulinoyl groups were selectively deprotected by treatment with hydrazine acetate in MeOH-tetrahydrofuran (THF), affording the tetrasaccharyl diol acceptor **23** (Scheme 3).

Next, the sialyl donor was prepared as shown in Scheme 4. To conjugate fluorescent dye with C9 amino group at the final step, we needed to convert C9 hydroxyl group into a trifluoroacetyl-protected amino group in advance. We chose a *tert*-buthoxycarbonyl (Boc) group as a protective group of the C5 amino group to retain the C9-trifluoroacetamido group during the retrieval of the C5 amino group. The Boc group was also expected to be insusceptible to the reaction conditions for replacing the C9 hydroxyl group with a trifluoroacetamido group. First, the C9 hydroxyl group in the *N*-Boc sialic acid derivative **24** [35] was protected with a Ts group, followed by azidation and one-pot reduction of the azido group and trifluoroacetylation of the resulting amine, producing the C9-trifluoroacetamido derivative **25**. Then, the Boc group was cleaved by treatment with trifluoroacetic acid, and the resulting amine was condensed with the *N*-succinimidyl carbonate **29**, which was prepared from alcohol **S8** (see Supporting Information) in situ, giving the carbamate derivative **26**. After acetylation of C4, 7, and 8 hydroxyl groups, demethylation, and cleavage of the TBS group, we performed intramolecular cyclization of compound **27** under the Mitsunobu condition to obtain the sialyl donor **28** without any intermolecular condensation. The use of bis(2-methoxyethyl) azodicarboxylate (DMEAD) [36,37] facilitated the separation of **28** from the hydrazine derivative produced from DMEAD due to its solubility in water.

Previously, we chemically synthesized various gangliosides and their analogs containing one or more sialic acid units. However, direct approaches to introduce sialic acid units to an oligosaccharide moiety have been avoided due to the uncertainties of stereoselectivity during sialylation. Here, we tried to introduce a sialic acid unit directly to the oligosaccharide using the fully α-selective sialylation method developed by our group. Although 3.0 equiv. of donor **28** was used to complete the reaction because of the self-decomposition of **28**, we coupled **28** with the tetrasaccharyl acceptor **23** at −50 °C, with a yield of 75%. After the removal of the DTBS group [32], we examined the opening of the 16-membered ring in the pentasaccharide **31**. By following the reported procedure [34], compound **31** was treated with zinc power in acetic acid. However, the reaction proceeded very slowly, which is probably due to the poor solubility of the reactant. When we diluted the reaction mixture with acetonitrile, the reaction proceeded more quickly but gradually produced an *N*-acetyl-8-hydroxy derivative resulting from O to N acetyl migration [38]. To obtain a single product, compound **31** was treated with zinc powder and an excess amount of acetic anhydride in AcOH-MeCN, successfully delivering **32** at a high yield. Compound **32** was advanced to hydrogenolysis, acetylation of the hydroxyl groups, cleavage of the MP group, and *N*-phenyltrifluoroacetimidoylation to provide the pentasaccharyl donor **35**.

Donor **35** was elongated with the GlcCer acceptor **17**, as mentioned above for the synthesis of fluorescent SSEA-3, producing the SSEA-4 glycolipid framework **36**, with a yield of 68%. Global deprotection and the introduction of the ATTO594 dye delivered the target ATTO594 SSEA-4 **2** (Scheme 5).

#### 2.1.5. Synthesis of the Fluorescent Globo-H Probe

First, the tetrasaccharyl acceptor **41** was prepared. The galactosyl donor **39**, protected with a levulinoyl group at the C2 hydroxyl group, was coupled with the trisaccharyl common acceptor **10** to yield the β-galactoside **40**. In this case, acid-washed 4 Å molecular sieves were used to suppress the formation of an orthoester. Next, cleavage of the levulinoyl group was achieved selectively to obtain the tetrasaccharyl acceptor **41**, with a high yield of 94%.

For furnishing α-fucoside, we used the synergic solvent effect of cyclopentyl methyl ether and CH_2_Cl_2_ [39]. The Fuc donor **42** [40] was coupled with the tetrasaccharyl acceptor **41** at 0 °C, building **43** with a yield of 82% as a single α-anomer. After manipulation of the protecting groups, the obtained donor **47** was elongated with the GlcCer acceptor **17**, giving the Globo-H framework **48**, with a yield of 75%. Global deprotection and the introduction of the ATTO594 dye delivered the target ATTO594 Globo-H **3** (Scheme 6).

### 2.2. Biophysical and Biochemical Evaluation

#### 2.2.1. Cold-Triton Solubility of the Fluorescent Probes in the PM

Putative raft-associated molecules are insoluble in detergent-resistant membrane (DRM) fractions [41]. Here, the cold-Triton solubilities of the chemically synthesized fluorescent probes in T24 human epithelial cells were examined by following the procedure we reported previously [8]. The cells in which the fluorescent probes were incorporated were treated with cold Triton X-100 and observed by epifluorescence microscopy (Figure 2). All of the synthesized analogs remained in the PM (=DRM), indicating that they retained their properties.

#### 2.2.2. Partitioning of the Fluorescent Probes between the Lo/Ld Phases

We examined the partitioning of the fluorescent analogs by using giant PM vesicles (GPMVs), which are largely depleted of the actin-based membrane skeleton, but are considered to contain the full complement of lipids and proteins of the native PM, separating them into the liquid-ordered (Lo, the putative raft phase) and liquid-disordered (Ld) phases [42,43,44]. Upon cooling the GPMVs, separation of the Lo and Ld phases could be observed (Figure 3).

GPMVs were prepared from RBL-2H3 cells at 10 °C, in which each of the fluorescent analogs and BodipyFL-PC, as Ld-like domain markers, were pre-incorporated. All of the synthesized analogs were preferentially partitioned into the Lo phase, indicating that they all retained their raftophilic properties.

## 3. Materials and Methods

### 3.1. Chemical Section

Detailed synthetic procedures and NMR spectra for all new compounds are described in the Appendix A.

### 3.2. Biological Evaluations

We carried out two biological evaluations following the reported procedures [8].

#### 3.2.1. Evaluation of Cold-Triton Solubility

For the epi-fluorescence imaging of fluorescent GSL analogs incorporated in the PM, the analogs were first dispersed in P-HBSS at a final concentration of 1.0 μM, and the dispersed solution was incubated with T24-cell PM at 22 °C for 15 min. Then, the cells were chilled on ice-water (2.8 °C), washed three times with pre-chilled P-HBSS, and incubated in pre-chilled P-HBSS containing 1% (v/v) Triton X-100 on ice-water for 15 min [41]. The Triton-treated cells were washed three times with cold P-HBSS, washed twice with pre-chilled PBS, fixed with 4% paraformaldehyde in PBS for 90 min, and observed under a Nikon Ts-2FL epi-fluorescence microscope equipped with a CMOS camera (DS-Qi2).

#### 3.2.2. Formation and Microscopic Observation of GPMVs

RBL-2H3 cells grown on a glass-bottom dish (Iwaki) were washed twice with P-HBSS (without phenol red and sodium bicarbonate; Nissui) and buffered at pH 7.4 with 2 mM PIPES. For the incorporation of the fluorescent ganglioside analogs into the PM, first, a 1-µL aliquot of the methanol stock solution of a fluorescent lipid analog (1 mM) was dried to make a thin film on the bottom of a vial and 100 µL P-HBSS was added. After vigorous vortexing for 1 min, the mixture was sonicated for 5 min. Subsequently, 0.1 mL of this suspension (final analog concentration of 10 µM) was overlaid on the cells in each glass-bottom dish, followed by incubation at 22 °C for 12 min. To mark the Ld-like domains, BodipyFL-PC was incorporated in the PM. To mark the Lo-like domains in GPMVs, the cells were incubated in 10 µM ATTO594-SSEA-3, ATTO594-SSEA-4, or ATTO594-Globo-H at 22 °C for 12 min.

After washing the cells three times with P-HBSS, membrane blebbing was induced by incubating the cells in 25 mM formaldehyde and 2 mM dithiothreitol in P-HBSS at 37 °C for 1 h. During this incubation period, numerous blebs were generated and detached from the cells to form GPMVs. The dish was moved onto the microscope stage of a TIRF microscope, based on an Olympus IX83, incubated at 20 °C for 15 min to let the GPMVs settle on the glass bottom, and cooled by circulating a chilled water-ethanol solution (−5 °C) so that the temperature of the top surface of the glass (facing the P-HBSS solution) stabilized at 10 °C. Under these conditions, the majority (>90%) of the GPMVs exhibited two coexisting domains: Ld-like domains preferentially concentrating BodipyFL-PC and Lo-like domains preferentially concentrating the GSL probe, as simultaneously observed by oblique angle illumination with 488 and 594-nm laser beams in our TIRF microscope. The focus was adjusted to collect the fluorescence signal emitted from the equatorial phase of the blebs. Images were recorded simultaneously in two emission channels (520 and 630 nm, or 520 and 670 nm) at a rate of 30 Hz. To estimate the partition coefficients of the various GSL analogs, only the first or second frame of the movie was used, which also minimized the photobleaching effect.

## 4. Conclusions

We have achieved the synthesis of fluorescent analogs of SSEA-3, SSEA-4, and Globo-H, which all retained the biophysical properties as raft molecules in the PM. To construct the trisaccharyl acceptor **10** and GSL frameworks **18**, **36**, and **48**, we applied *p*-*tert*-butyl-substituted protective groups (TBBzld and TBBz) for the first time toward the chemical synthesis of the complex glycoconjugates. For the synthesis of the SSEA-4 analog, we also achieved the direct sialylation of the SSEA-3 tetrasaccharide moiety, which enabled the synthesis of various biofunctionalized molecules, including sialic acid(s).

Given the high pluripotency of hESCs and hiPSCs, and the high proliferation capacity of cancer cell tissues, it has been difficult to understand the behavior of these molecules in the PM of these cells. The fluorescent probes we have developed in this study will accelerate the study of microdomains during cell differentiation and cell malignancy.

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
