# Peer review of "Development of Fluorescently Labeled SSEA-3, SSEA-4, and Globo-H Glycosphingolipids for Elucidating Molecular Interactions in the Cell Membrane"

_ijms, 2019, doi:10.3390/ijms20246187_

Round 1

Reviewer 1 Report

This article describes the syntheses of fluorescent analogues of glycosphingolipids belonging to the globo-series, specifically stage-specific embryonic antigen 3 (SSEA-3), SSEA-4 and Globo-H. These analogs have been synthesized using the GlcCer cassette approach described by Imamura and Kiso in 2009, employing a common trisaccharide unit.

The interest of the synthesis is justified by the biological properties of reference GSL, whose behaviour on the cellular surface during cell differentiation and cell malignancy could be studied using the compounds obtained in this work. The study of the biophysical properties of the fluorescents glycosphingolipids obtained here effectively shows that they behave like the corresponding native products, which confirms their potential use to elucidate the molecular interactions in the cell membrane.

The global synthesis has been well planned, following a convergent approach with a good bibliographic justification of the methodologies employed therein. Thus, in the discussion of the synthesis, both the positive aspects of it are highlighted (such as the use of Troc as a protecting group, the choice of protecting groups for the sialyl donor or the α-selective sialylation developed by the group), as well as its limitations (obtaining the orthoester in the synthesis of disaccharide 6). The supporting information is complete with a good structural characterization of the obtained products.

Some aspects, which should be taken into consideration, are indicated below.

The introduction of the work is written clearly and directly, although the references cited in this section are not very updated. It would be convenient to introduce some recent publications on glycosphingolipids in general, as well as some related with their function and their disposition in the cell membrane in references 1-5 (e.g. “Glycophingolipids functions”, Cold Spring Harb Perspect Biol 2011;3:a004788). In this sense, it is also advisable to include the citation of the book “Essentials of Glycobiology” (3rd edition, 2017. Chapter 11: “Glycosphingolipids”). For the synthesis of tetrasaccharide 40 an acid-washed 4Å molecular sieves was used to avoid orthoester formation. Why was this procedure not used in the synthesis of product 6 that suffers from the same problem? An explanation or a description on the glycosylation for obtaining 6 under these conditions should be included in the manuscript. The standard abbreviation for trifluoroacetic acid is TFA and not TFAcOH. Please, change this denomination in the whole document and Supporting information.

Taking into account the above considerations, the paper should be published in Int. J. Mol. Sci. after minor revisions.

Author Response

We are grateful for the critical reading and useful suggestions that helped to improve our manuscript. As indicated in the following responses, we have taken all comments and suggestions into account in the revised manuscript.

[Comment #1]

The introduction of the work is written clearly and directly, although the references cited in this section are not very updated. It would be convenient to introduce some recent publications on glycosphingolipids in general, as well as some related with their function and their disposition in the cell membrane in references 1-5 (e.g. “Glycophingolipids functions”, Cold Spring Harb Perspect Biol 2011;3:a004788). In this sense, it is also advisable to include the citation of the book “Essentials of Glycobiology” (3rd edition, 2017. Chapter 11: “Glycosphingolipids”).

[Response]

Thank you very much for your suggestions. We have added the suggested references in the revised manuscript.

[Comment #2]

For the synthesis of tetrasaccharide 40 an acid-washed 4Å molecular sieves was used to avoid orthoester formation. Why was this procedure not used in the synthesis of product 6 that suffers from the same problem? An explanation or a description on the glycosylation for obtaining 6 under these conditions should be included in the manuscript.

[Response]

Thank you very much for the comment. The glycosidation reaction to obtaine 6 is α-glycosidation while teterasaccharide 40 was produced by β-glycosidation, which produces orthoeseter via side reaction path. So, acid-washed 4A molecular sieves were not used in the synthesis of 6.  This explanation might not fit with the flow of the main text since it is common in carbohydrate chemistry.

[Comment #3]

The standard abbreviation for trifluoroacetic acid is TFA and not TFAcOH. Please, change this denomination in the whole document and Supporting information.

[Response]

Thank you for the suggestion. We have replaced with TFAOH in the revised manuscript and SI.

Reviewer 2 Report

Comments: 1. References: This manuscript suffers from lack of updated references . For example ref 2-4 are outdated and the authors should provide more recent publications showing the functional role of GSL in health and disease. 2. GSL are not only localized with the plasma membrane but also cytoplasm and Golgi-the authors should refer to this along with appropriate references. 3.Scheme 2 fails to identify tetrasccharide 16 and SSEA-3 18. Also how physiologic GLcCer is converted to its soluble from 17 needs to be shown for the benefit of general readership. 4. Please elaborate on “ fluorescent tagged GSL behave like natural substances “

Author Response

We are grateful for the critical reading and useful suggestions that helped to improve our manuscript. As indicated in the following responses, we have taken all comments and suggestions into account in the revised manuscript.

[Comment #1]

References: This manuscript suffers from lack of updated references. For example ref 2-4 are outdated and the authors should provide more recent publications showing the functional role of GSL in health and disease.

[Response]

Thank you for the suggestion. By following the advice from the reviewer 1, we have updated the references regarding to GSL functions. 

[Comment #2]

GSL are not only localized with the plasma membrane but also cytoplasm and Golgi-the authors should refer to this along with appropriate references.

[Response]

We appreciate you for the comment. We have modified the relevant phrase with reference 1 (suggested by the reviewer 1) as follows.

‘Glycosphingolipids (GSLs) are lipid molecules that are present in the plasma membrane (PM) as well as in cytoplasm and Golgi complex, and contain at least one monosaccharide residue with ceramide lipid that consists of a sphingoid base with a fatty acid amide at the C2 amine [1]’

[Comment #3]

Scheme 2 fails to identify tetrasccharide 16 and SSEA-3 18. Also how physiologic GLcCer is converted to its soluble from 17 needs to be shown for the benefit of general readership.

[Response]

Thank you for the comment. In Scheme 2, the chemical structure of 16 has been indicated based on 12. To minimize the size of scheme showing chemical reactions, we usually minimize the number of the full chemical structures of huge molecule by showing only substituent change for derivatized molecules from the parent intermediate displayed with full structures. 

We have put a brief explanation about how GlcCer was converted into the soluble form 17 as follows.

‘To construct the glycolipid framework of SSEA-3, we applied a highly soluble GlcCer acceptor 17 as a coupling partner for the tetrasaccharide donor 16, which was chemically synthesized by glycosylation of 3-p-tert-butylbenzoyl (TBBz)-protected Cer acceptor with fully protected Glc donor [32]. In GlcCer 17, bulky TBBz groups were installed as self-aggregation inhibitor, making 17 highly soluble in organic solvents.

[Comment #4]

Please elaborate on “ fluorescent tagged GSL behave like natural substances “

[Response]

Thank you for the suggestion. We have revised the phrase as follows.

‘……..which all retained the biophysical properties as raft molecules in the PM’

Reviewer 3 Report

Nicely done and presented very elaborate work.

Author Response

We are grateful for the critical reading and the positive comment.